# SFRP4 Expression Is Linked to Immune-Driven Fibrotic Conditions, Correlates with Skin and Lung Fibrosis in SSc and a Potential EMT Biomarker

**DOI:** 10.3390/jcm10245820

**Published:** 2021-12-13

**Authors:** Ilaria Tinazzi, Panji Mulipa, Chiara Colato, Giuseppina Abignano, Andrea Ballarin, Domenico Biasi, Paul Emery, Rebecca L. Ross, Francesco Del Galdo

**Affiliations:** 1Leeds Institute of Rheumatic and Musculoskeletal Medicine, University of Leeds, Leeds LS7 9TF, UK; ilariatinazzi@yahoo.it (I.T.); P.q.mulipa@leeds.ac.uk (P.M.); G.Abignano@leeds.ac.uk (G.A.); P.Emery@leeds.ac.uk (P.E.); 2Unit of Rheumatology, IRCSS Ospedale Sacro Cuore-Don Calabria, 37024 Verona, Italy; 3Section of Pathology, Department of Diagnostics and Public Health, University of Verona, 37134 Verona, Italy; chiara.colato@aovr.veneto.it; 4NIHR Leeds Biomedical Research Centre, Leeds Teaching Hospitals NHS Trust, Leeds LS7 4SA, UK; 5Department of Surgery, University Hospital of Verona, 37126 Verona, Italy; andrea.ballarin@aovr.veneto.it; 6Unit of Rheumatology, University Hospital of Verona, 37126 Verona, Italy; domenico.biasi@univr.it

**Keywords:** systemic sclerosis, SFRP4, GVHD, fibrosis, wnt, caveolin-1

## Abstract

Secreted Frizzled Receptor Protein 4 (SFRP4) has been shown to be increased in Scleroderma (SSc). To determine its role in immune-driven fibrosis, we analysed SSc and sclerotic Chronic Graft Versus Host Disease (sclGVHD) biosamples; skin biopsies (*n* = 24) from chronic GVHD patients (8 with and 5 without sclGVHD), 8 from SSc and 3 healthy controls (HC) were analysed by immunofluorescence (IF) and SSc patient sera (*n* = 77) assessed by ELISA. Epithelial cell lines used for in vitro Epithelial-Mesenchymal-Transition (EMT) assays and analysed by Western Blot, RT-PCR and immunofluorescence. SclGVHD skin biopsies resembled pathologic features of SSc. IF of fibrotic skin biopsies indicated the major source of SFRP4 expression were dermal fibroblasts, melanocytes and vimentin positive/caveolin-1 negative cells in the basal layer of the epidermis. In vitro studies showed increased vimentin and SFRP4 expression accompanied with decreased caveolin-1 expression during TGFβ-induced EMT. Additionally, SFRP4 serum concentration correlated with severity of lung and skin fibrosis in SSc. In conclusion, SFRP4 expression is increased during skin fibrosis in two different immune-driven conditions, and during an in vitro EMT model. Its serum levels correlate with skin and lung fibrosis in SSc and may function as biomarker of EMT. Further studies are warranted to elucidate the role of SFRP4 in EMT within the pathogenesis of tissue fibrosis.

## 1. Introduction

Systemic Sclerosis (SSc) is characterized by excessive deposition of collagen and other connective tissue macromolecules in skin and multiple internal organs, prominent alterations in the microvasculature and immunologic abnormalities [1,2]. The clinical course of SSc is extremely heterogeneous regarding both the predominance of the fibrotic vs. the vasculopathic process and in the nature and severity of organ involvement. Biosamples reflect this heterogeneity and for this reason, identification of disease mechanisms from biosampling is extremely complicated [3]. However, it is clear that in every patient, vasculopathic and fibrotic phenotypes are also present with autoimmunity. Microarray studies have led to a proposed molecular classification of the disease implying that disease heterogeneity, which is clinically poorly predictable, is reflected at molecular level [4].

The pathophysiology of lung and skin fibrosis remains to be discovered; however, the current concept suggests injury-induced activation of fibroblasts and epithelial cells and impaired epithelial-mesenchymal crosstalk [5]. There is a strong link between activation of signalling pathways, such as the Wnt pathway, and an abnormal repair response as well as representing a hallmark for SSc fibrotic phenotype [6,7]. Secreted frizzled-related proteins (SFRPs) bind Wnt ligands and have a complex effect on Wnt gradient formation, as well as extending the ligand signalling range [8,9]. Previous studies have indeed implicated abnormal SFRPs expression, including increased SFRP4 expression, in SSc and in a mouse model of SSc [4,10,11,12,13,14,15]. In particular, single cell RNAseq revealed increased SFRP4 in a distinct subgroup of fibroblasts isolated from SSc-interstitial lung disease (ILD) biosamples, which were hypothesised to be progenitors of myofibroblasts [16]. Furthermore, RNAseq of SSc skin biopsies highlighted SFRP4 increased expression relative to healthy controls, especially in diffuse SSc (dcSSc) [17].

The Chronic Graft Versus Host Disease (GVHD) class of conditions are a very useful model to study the immunopathogenesis of tissue fibrosis, since patients with the same immune trigger of a HLA matched allogeneic Stem Cell transplant develop a chronic rejection of the bone marrow against the host, with or without skin fibrosis [18,19]. The fibrotic variant, also called sclerotic GVHD (sclGVHD) resembles many features of Scleroderma, and mouse models of sclGVHD have previously been used to understand the pathogenesis of SSc [20,21,22]. Here we set out to determine whether increased SFRP4 plays a general role in immune-driven fibrosis, by comparing SSc and sclGVHD samples and determine the source of its increased expression.

## 2. Materials and Methods

### 2.1. Patients

Ethical approval was obtained for this study from the Ethical Committee (EC) of Verona University (Verona, Italy) (study number VR409). A 6-mm punch skin biopsy was taken from a total of 24 individuals: 5 patients with diffuse cutaneous SSc (dcSSc) and 3 patients with limited cutaneous SSc (lcSSc) classified according to LeRoy et al. [23], 8 patients affected by sclGVHD, 5 patients affected by cGVHD (3 with lichenoid and 2 dyscromic phenotype) and 3 healthy volunteers. Patients were excluded from the study if receiving prednisone ≥20 mg during the 4 weeks prior to the biopsy. For sclGVHD and SSc patients, specimens were taken from areas were the modified Rodnan Skin Score (mRSS) [24] was 2. Sera of 77 consecutive patients (62 female, mean age 57.5 ± 14.3 years, mean disease duration 15.2 ± 13.6 years) were collected and stored at −80 °C until use. All scleroderma patients fulfilled the American College of Rheumatology (ACR) classification criteria for SSc [25].

### 2.2. Histology Studies

Formalin-fixed skin specimens were embedded in paraffin. Cut sections (3 µm) were stained with either Hematoxylin-Eosin (H&E) or Trichrome blue. Dermal fibrosis score (DFS), which reflects the percentage of sclerosis in the full thickness of dermis by quintiles was scored by an expert pathologist blinded to clinical details in grade 1, 25%, grade 2, 25–50%, grade 3, 50–75%, grade 4 pandermal sclerosis, grade 5 pandermal sclerosis with extension into the hypodermis as described [26]. We also used integrated density of autofluorescence of ECM by fluorescence microscopy to quantify the amount of skin fibrosis similar to what has been described by Busquets et al. [27]. Luminal ratio of vessels within papillary dermis (PD) in skin biopsies of HC, SSc, sclGVHD and cGVHD patients was calculated dividing the total area of vessel to the lumen area of the same vessel and quantified by image J software (NIH). Immunohistochemistry analysis (IHC) was performed on paraffin embedded sections. Slides were heated, deparaffinized and hydrated immediately before commencing immunostaining. The sections were washed in PBS buffer for 2 min before incubation with 3% H_2_O_2_ (in methanol) for 25 min at room temperature to block endogenous peroxide activity before 3% bovine serum albumin in PBS for 60 min at room temperature to block nonspecific binding. Sections were counterstained in Harris’s hematoxylin for 45 s, dehydrated, cleared and mounted in DPX. Slides were examined using an Olympus BX50 with MicroFire (Optronics) microscope connected to a Micro Brightfield MBF camera and using Stereo Investigator software; endothelial cells were counted at 40× magnification in 8 random fields.

### 2.3. Immunofluorescence

Immunofluorescent studies were performed on paraffin embedded sections. Slides were heated, deparaffinized and then hydrated, blocked by 5% BSA 20 min and incubated with mouse monoclonal anti-vimentin antibody at 1:200 (Abcam, Cambridge, UK), rabbit polyclonal to sFRP4 antibody at 1:25 (Abcam), mouse polyclonal anti-caveolin-1 antibody at 1:500 (BD, Wokingham, UK) or rabbit polyclonal affinity-purified anti-caveolin-1 antibody at 1:500 (Santa Cruz Biotechnology, Dallas, TX, USA), Melanasome marker (HMB45 1:200; DAKO, Santa Clara, CA. USA) in PBS overnight at 4 °C. Sections were washed with PBS. To detect primary antibodies, sections were incubated with fluorescein isothiocyanate–conjugated anti–rabbit-IgG at 1:200 or Cy3-conjugated anti–mouse-IgG 1:200 (Abcam). Diamidino-2-phenylindole dihydrochloride (1 g/mL) was used to stain nuclei. Confocal Laser Scanning Microscopy was performed using an Olympus BX61 upright fluorescence microscope with 20× dry and 40× oil immersion objectives. Images were captured using EZ-C1 3.80 software (Nikon, Düsseldorf, Germany), the same software was used in balancing signal strength. The 3 channels were recorded simultaneously if no signal overlap could be detected. In the case of strong FITC labeling, sequential images were acquired at more restrictive wave-lengths to prevent possible breakthrough of the FITC signal into the red channel. Both acquisition modes resulted in the same images. The image was scanned 4-fold to separate signal from noise. Panels were assembled using Photoshop software (Adobe Systems, San Jose, CA, USA) without any RGB modification.

### 2.4. Tissue Culture and Functional In Vitro Analysis

A549 epithelial cells were cultured in Dulbecco’s modified Eagle’s medium (DMEM; Invitrogen, Carlsbad, CA, USA) supplemented with 10% fetal bovine serum (FBS) (Invitrogen), antibiotics, and glutamine. Cells were starved for 18 h in 0.5% FBS and subsequently treated for 72 h with or without 10 ng/mL human recombinant TGF-β (Sigma, St. Louis, MO, USA). Primary healthy skin epithelial cells were cultured in Keratinocyte Growth Medium 2 (Promocell, Heidelberg, Germany). Cells were starved for 24 h in CaCl_2_ only media and subsequently treated for 72 h with or without 10 ng/mL TGF-β. RNA from A549 epithelial cells was extracted employing Qiagen RNAeasy columns, according to manufacturer’s instructions while RNA from primary healthy skin epithelial cells was extracted using the Zymo quick RNA mini prep kit, according to manufacturers’ instructions (Zymo research, Irvine, CA, USA). First-strand cDNA was synthesized from A549 epithelial cell RNA using the SuperScript III chemistry, according to manufacturers’ instructions while cDNA was synthesised from primary epithelial cells using the High-Capacity cDNA Reverse Transcription kit (ThermoFisher Scientific, Waltham, MA, USA). Quantitative PCR using SYBR Green Master Mix chemistry was performed according to the manufacturer’s instructions, following a standard amplification protocol, with an ABI Prism 7700 Sequence Detection System (Applied Biosystems, Foster City, CA, USA). The differences in the number of mRNA copies in each PCR were corrected for human ribosomal 18 s RNA or GAPDH endogenous control transcript levels. Primer sequences used in qPCR analyses were as follows: 18S ribosomal RNA forward 5′-GTA ACC CGT TGA ACC CCA TT-3′, reverse 5′-CCA ATA ATC GGT AGT AGC G-3′. GAPDH forward 5′-ACC CAC TCC TCC ACC ACC TTT GA-3′, reverse 5′-CTG TTG CTG TAG CCA AAT TCG T-3′. Caveolin-1 forward 5′-CGA CCC TAA ACA CCT CAA CGA-3′, reverse 5′-TCC CTT CTG GTT CTG CA-3′. Col1a1 forward 5′-GCT CCG ACC CTG CCG ATG TG-3′, reverse 5′-CAT CAG GCG CAG GAA GGT CAG C-3′. E-Cadherin forward 5′-AAA TCT GAA AGC GGC TGA TAC TG-3′, reverse 5′-CGG AAC CGC TTC CTT CAT AG-3′. N-Cadherin forward 5′-GCG TCT GTA GAG GCT TCT GG-3′, reverse 5′-GCC ACT TGC CAC TTT TCC TG-3′. Snail forward 5′-TTC AAC TGC AAA TAC TGC AAC AAG-3′, reverse 5′-TGT GGC TTC GGA TGT GCA T-3′. SFRP4 forward 5′-CGA TCG GTG CAA GTG TAA A-3′, reverse 5′-GAC TTG AGT TCG AGG GAT GG-3′. Vimentin forward 5′-CCT GTG AAG TGG ATG CCC TTA-3′, reverse 5′-AGC TTC AAC GGC AAA GTT CTC T-3′.

Intracellular proteins in whole cell lysates were obtained by lysing the cells in 6-well plates and 50 µg of total protein was subjected to western blotting as described previously [28]. Antibodies used: SFRP4 polyclonal rabbit antibody (Abcam) at a 1:500 dilution and rabbit anti-human actin antibodies (Sigma) at 1:1000 dilution in 5% milk-Tris buffered saline-Tween (TBST). E-cadherin polyclonal rabbit antibody (Santa Cruz Biotechnology, sc7870) at 1:1000 dilution and Vimentin monoclonal antibody (Invitrogen, MA5-14564) at 1:200 dilution. B-actin monoclonal mouse antibody (Sigma, A5441) at 1:5000 dilution. Horseradish peroxidase–conjugated secondary antibodies (Cell Signaling technology, London, UK) were used at 1:5000 dilution.

For immunofluorescence Phalloidin staining, cells were paraformaldehyde fixed and stained with Alexa Fluor^TM^ 488 Phalloidin (Thermofisher Scientific) according to manufacturers’ instructions. Slides were mounted with VectaShield Antifade mounting medium with DAPI (Vector Laboratories, Burlingame, CA, USA) and visualized using ×10 magnification on the EVOS™ FL Imaging System (Thermofisher Scientific).

### 2.5. ELISA for SFRP4

The concentration of SFRP4 present in the sera of SSc patients was determined using a commercially available sandwich immunoassay (SFRP4 ELISA, USCNK life science Inc., Wuhan, China) as per manufacturer’s instruction. In summary, a standard curve with a detection range between 1000 pg/mL and 15.6 pg/mL was prepared using the supplied recombinant protein and standard diluent. Sera samples and protein standards were added to the pre-coated 96 well ELISA plate and incubated for 2 h at 37 °C. Detection reagent A was added for 1 h at 37 °C and the plate washed 3 times using the supplied wash buffer. Detection reagent B was added for 30 min at 37 °C and followed by wash buffer 3 times. Substrate solution was added and catalysis proceeded for a period of up to 5 min at RT. The reaction was stopped using stop solution and the absorbance of each well was read at 450 nm using an OpsysMR plate reader (Dynex Technologies, Chantilly, VA, USA).

### 2.6. Statistical Analysis

Continuous variables were expressed as mean ± standard error of the mean (SEM) and categorical data as number and percentage. The Wilcoxon signed rank test for paired samples and the Mann-Whitney U test for unpaired samples were used. Correlations were calculated using Spearman’s rank correlation test. Paired student t tests were performed for qPCR data. *p* value < 0.05 was considered statistically significant. Statistical analysis was performed using GraphPad Prism software, version 5.0 (San Diego, CA, USA).

## 3. Results

### 3.1. SSc and sclGVHD Share Similar Histologic Features

The demographics and clinical features of all patients that underwent skin biopsy are summarized in Table 1. Patients with (sclGVHD) or without (cGVHD) fibrotic skin involvement were matched for age, gender, age at Hematopoietic Stem Cell Transplant (HSCT) and time between HSCT and cGVHD onset. Both SSc patients and sclGVHD were considered affected by fibrotic process at the blinded analysis of an expert pathologist. Representative Masson trichrome staining of healthy control (HC), cGVHD, sclGVHD and SSc skin are shown in Figure 1A. DFS values demonstrated that 75% of SSc patients and 87.5% of sclGVHD patients had a percentage of sclerosis in the full thickness of dermis over 50% (Table 1). The overall score was 3.25 ± 1.035 (*p* = 0.0121 vs. HC) and 3.75 ± 1.035 (*p* = 0.0061 vs. HC) in SSc and sclGVHD, respectively (Figure 1B).

Semiquantitative assessment of extracellular matrix (ECM) autofluorescence indicated that sclGVHD had an increased ECM content both in papillary (fold = 1.5, *p* = 0.0008) and reticular dermis (RD) (fold = 1.5, *p* = 0.0007) compared to HC, similar to what is observed in SSc (Figure 1C,D). On the contrary cGVHD skin biopsies showed no significant difference in ECM autofluorescence when compared to HC (fold = 1.02 and 1.16, *p* > 0.05).

By measuring the vessel luminal ratio within the papillary dermis of sclGVHD biopsies, we saw a reduction (32%, *p* < 0.05) similar to the one observed in SSc (41%, *p* < 0.05) relative to HC and cGVHD samples (Figure 1E).

### 3.2. SFRP4 Expression Is Increased in sclGVHD Epidermis and the Germinal Layer

Immunofluorescence studies showed increased expression of SFRP4 in sclGVHD skin biopsies compared to cGVHD (Figure 2A), as previously seen in SSc [4,10,11,12,13,14,15,16,17]. Positive cells included both dermal fibroblasts and cells within the basal level of the epidermis (Figure 2A). Double immunofluorescence studies on SSc skin biopsy sections indicated that SFRP4 positive cells in the basal layer of the epidermis lacked Caveolin-1 and expressed Vimentin (Figure 2B,C). Indeed, all Vimentin positive cells in the basal layer of the epidermis did not express caveolin on their membrane (Figure 2D). Some of these cells were melanocytes, whereas some others did not express any melanocyte marker (HMB-45), indicating that they were of epidermal origin (Figure 2E).

### 3.3. SFRP4 Is Induced during In Vitro TGFβ Driven Epithelial-to-Mesenchymal Transition (EMT)

The observation of SFRP4 and Vimentin co-expression in the epidermis of SSc skin led us to assess their expression during an in vitro EMT model [29]. A549 lung carcinoma epithelial cells were induced to EMT by 72 h stimulation with TGFβ. As expected, TGFβ induced EMT in A549 cells, as indicated by both increase of Col1A1 expression by 27.8 fold and increase in Snail mRNA levels by 4.6 fold, as expected (Figure 3A). This transition was accompanied by a decrease in Caveolin-1 expression by 92% and an increase of SFRP4 both at mRNA and protein level (1.26 and 1.86-fold, respectively) (Figure 3A,B). The EMT model was also used to assess what happens in primary healthy skin epithelial cells (Figure 3C–E). Similarly, Col1A1, SFRP4 and Snail mRNA expression were increased, which was accompanied with increased N-Cadherin and Vimentin mRNA expression and decreased E-Cadherin and Caveolin-1 expression (Figure 3C). Western blot analysis showed TGFβ-reduced E-Cadherin and increased Vimentin protein expression (Figure 3D). Phalloidin immunofluorescence staining shows the changes in cell morphology by TGF treatment (Figure 3E).

### 3.4. SFRP4 Serum Concentration as Putative Biomarker of Lung or Skin Fibrosis in SSc

Sera from 77 SSc patients were assayed by ELISA for SFRP4. Clinical features of the 77 SSc patients are reported in Table 2. Thirty-nine patients suffered from lcSSc, 38 from dcSSc; 39 patients were anti-centromere (ACA) positive, 30 anti-topoisomerase I positive. Mean SFRP4 serum concentration was 32.07 ng/mL ± 11.94 (Figure 4A). The concentration of SFRP4 was higher in patients with dcSSc vs lcSSc (35.52 ± 12.49 vs. 28.71 ± 10.46, *p* = 0.0093), in patients anti-topoisomerase I (Scl-70) positive vs negative (36.19 ± 13.77 vs. 29.44 ± 19.89, *p* = 0.0145) and ACA negative vs positive (36.55 ± 12.94 vs. 27.7 ± 9.06, *p* = 0.0006) (Figure 4A).

SFRP4 concentration showed an inverse correlation with forced vital capacity (FVC%) (r = −0.27; *p* = 0.0171) and diffusion lung capacity of carbon monoxide (DLCO%) % of predicted value (r = −0.29; *p* = 0.0109) (Figure 4B,C). Univariate analysis showed that mRSS did not correlate with SFRP4 values in the overall population. Nevertheless, given the correlation with DLCO and FVC %, we set out to analyse the correlation of SFRP4 and mRSS in the patients with normal lung function (FVC and DLCO ≥ 70%). Indeed, in this subpopulation, we observed a significant correlation of circulating levels of SFRP4 and mRSS (r = 0.40; *p* = 0.0061) (Figure 4D).

## 4. Discussion

The identification of key molecular events in the pathogenesis of SSc is hampered both by the heterogeneity of the condition and by the concurrent presence of several pathogenetic components, including the autoimmune activation and the fibroproliferative vasculopathy. Recently, a productive translational medicine approach has consisted in selecting “extreme phenotypes” for the identification of key molecular events that could be involved in the pathogenesis of the given condition. A similar approach consists in comparing conditions with a similar clinical phenotype and identifying putative molecular events by metanalysis of genes and or proteins with similar pattern of expression. We have already applied this paradigm effectively by comparing the secretome of SSc and Nephrogenic systemic fibrosis fibroblasts [30]. Here we aimed to identify the molecular events that may be central in the pathogenesis of tissue fibrosis by comparing two different conditions with a similar skin phenotype, SSc and sclGVHD. Gene array studies have already identified a signature of genes that are consistently up-regulated in SSc. Correlation between the relative expression of these genes and disease phenotype has led to attempt a molecular classification of SSc [4].

Chronic GVHD is an immune-driven condition that can express clinically with or without skin fibrosis. For this reason we hypothesised that genes of the Scleroderma signature that showed a similar pattern of expression in the fibrotic GVHD phenotype, and are not involved in the non-fibrotic phenotype could have a potentially important role in tissue fibrosis caused by allo- or auto-immune trigger. Immunohistochemistry studies of cGVHD and sclGVHD skin biopsies are insufficient [31]. Here we analysed the presence and amount of skin fibrosis in these two variants of cGVHD and measured the amount of vasculopathy that could be detected by immunohistochemistry. Consistent with previous data, we observed an increased ECM content in sclGVHD [31]. To better quantify the amount of ECM in sclGVHD skin biopsies we set out to determine the amount of ECM by fluorescence microscopy. A similar approach, but employing indirect immunofluorescence of ECM antigens, has been already proposed by Busquets et al. [27]. Indeed, here we found that by direct fluorescence we could quantify the increased amount of ECM in both the papillary and reticular dermis of sclGVHD and SSc compared to HC and cGVHD. Furthermore, analysis of vessel lumen ratio showed that, despite the preserved number of vessels, consistent with previous findings [31], we could observe a reduction of vessel lumen ratio in scl-GVHD compared to HC and cGVHD, similar to the one observed in SSc. Altogether, the presence of increased ECM deposition and fibroproliferative vasculopathy further supported the use of sclGVHD skin biopsies for the identification of putative molecules involved in the pathogenesis of skin fibrosis in SSc.

Previous literature has indicated that SFRP4 shows increased expression in SSc affected vs not affected skin and lung both at mRNA level and protein level and in an animal model of skin fibrosis [4,10,11,12,13,14,15,16,17]. Here we set out to determine whether, by employing sclGVHD and cGVHD skin biopsies we could validate the role of SFRP4 in the profibrotic process. Indeed, immunofluorescence studies showed that sclGVHD skin biopsies have increased expression of SFRP4, compared to non-sclerotic GVHD. Similar to SSc, besides the already described increased expression of SFRP4 in dermal fibroblasts [12,14,16], cells from the basal layer of the epidermis also showed increased expression of SFRP4. A careful analysis of published studies indicated that Bayle et al. observed similar findings, though this was not commented or described in detail in the study [15]. Double immunofluorescence studies indicated that the source of increased expression of SFRP4 in the germinal layer of the epidermis were both melanocytes and other putative epithelial cells that expressed Vimentin and lost the expression of Caveolin-1. While the expression of Vimentin in epithelial cells is a clear marker of mesenchymal cells, we could not prove that the EMT process was actually happening in the affected skin. Specifically, it is possible that the vimentin-positive cells could be Langerhans cells. To test the hypothesis that a putative EMT of epithelial cells would involve the up-regulation of SFRP4 and down-regulation of Caveolin-1 expression, we tested in vitro the expression of these genes in a TGF-induced EMT cell model using epithelial cells [29]. Indeed, we found that during EMT, Caveolin-1 expression was clearly reduced and SFRP4 mRNA and protein levels were induced in A569 epithelial cells, which was linked with increased collagen and Snail expression. In healthy primary epithelial cells, increased SFRP4 expression was also associated with TGF-induced EMT, shown by increased expression of mesenchymal markers collagen, Vimentin, Snail and N-cadherin, decreased epithelial marker E-cadherin and altered cuboidal to elongated cell morphology. Further studies are needed to determine the putative role of SFRP4 in EMT and whether this is involved in myofibroblast differentiation, since previously SFRP4 has been shown to be a marker of this subset of SSc fibroblasts [32]. Previous observations show strong correlation between SFRP4 and other Wnt-related gene expression, including WIF1 and WNT2, in SSc skin and fibrosis [15,16,32,33]; thus, it is likely that SFRP4 is a marker for aberrant Wnt signalling, and warrants further research. Additional studies are needed to determine whether SFRP4 and EMT is a pathologic step in fibrosis of SSc and sclGVHD. Although the in vitro and biomarker data do suggest that TGF may induce SFRP4 in epithelial cells, another possible origin of SFPR4 staining in SSc may be Langerhans cells. While their relative amount would not justify an increase in serum concentration detectable in SSc patients, further experiments should be designed to assess this possibility. Furthermore, the observation that melanocytes are one of the sources of SFRP4 increased expression in SSc surely deserves further studies, given the well described alterations of skin pigmentation described both in SSc and sclGVHD [34].

Beyond determining the putative role of SFRP4 in the pathogenesis of skin fibrosis, our study suggests that SFRP4 secretion may function as a biomarker of EMT. We observed higher SFRP4 levels in patients with anti-topoisomerase I antibodies (a marker of diffuse skin involvement) and in general in patients with dcSSc, which agrees with previous observations of increased SFRP4 mRNA expression in dcSSc compared to lcSSc relative to HC [17]. Additionally, SFRP4 levels showed an inverse correlation with lung function and a direct correlation with skin involvement in patients that did not have lung involvement. Altogether, we believe that the data we show here strongly warrant further studies to determine the role of SFRP4 in the pathology of skin and lung fibrosis.

## Figures and Tables

**Figure 1 jcm-10-05820-f001:**
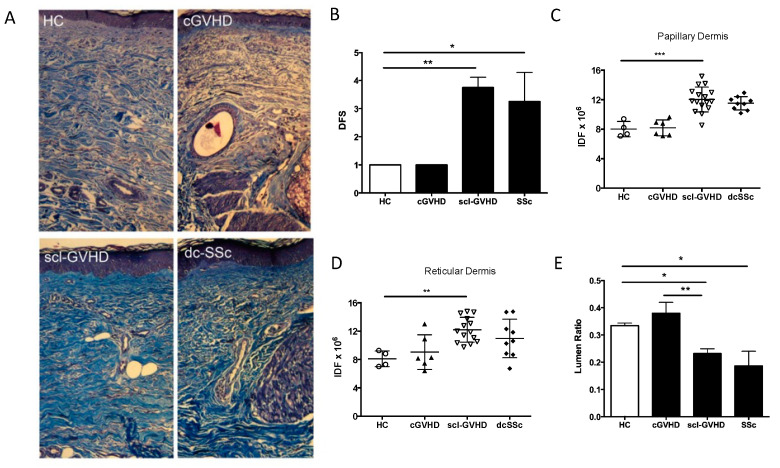
SSc and sclGVHD share similar histologic features. (**A**) Representative masson trichrome skin images selected from the study population of an healthy control, chronic GVHD (without fibrosis; cGVHD), sclGVHD (fibrotic phenotype) and SSc (dc; diffuse phenotype). Original magnification is 200× for each of the panels. (**B**) Dermal Fibrosis Score (DFS) of the 24 skin biopsies grouped for diagnosis. White represents healthy control (HC). Bars represent mean and Standard Error values. (**C**,**D**) Quantification of tissue fibrosis by integrated density of Extracellular matrix autofluorescence (IDF). Each dot represents the mean IDF of two low magnification fields in the papillary or reticular derma. (**E**) Mean value of luminal area of 10 vessels in papillary derma calculated by Image J software. White represents HC. Bars represent Mean and Standard Error. * = *p* < 0.05; ** = *p* < 0.001; *** = *p* < 0.0001.

**Figure 2 jcm-10-05820-f002:**
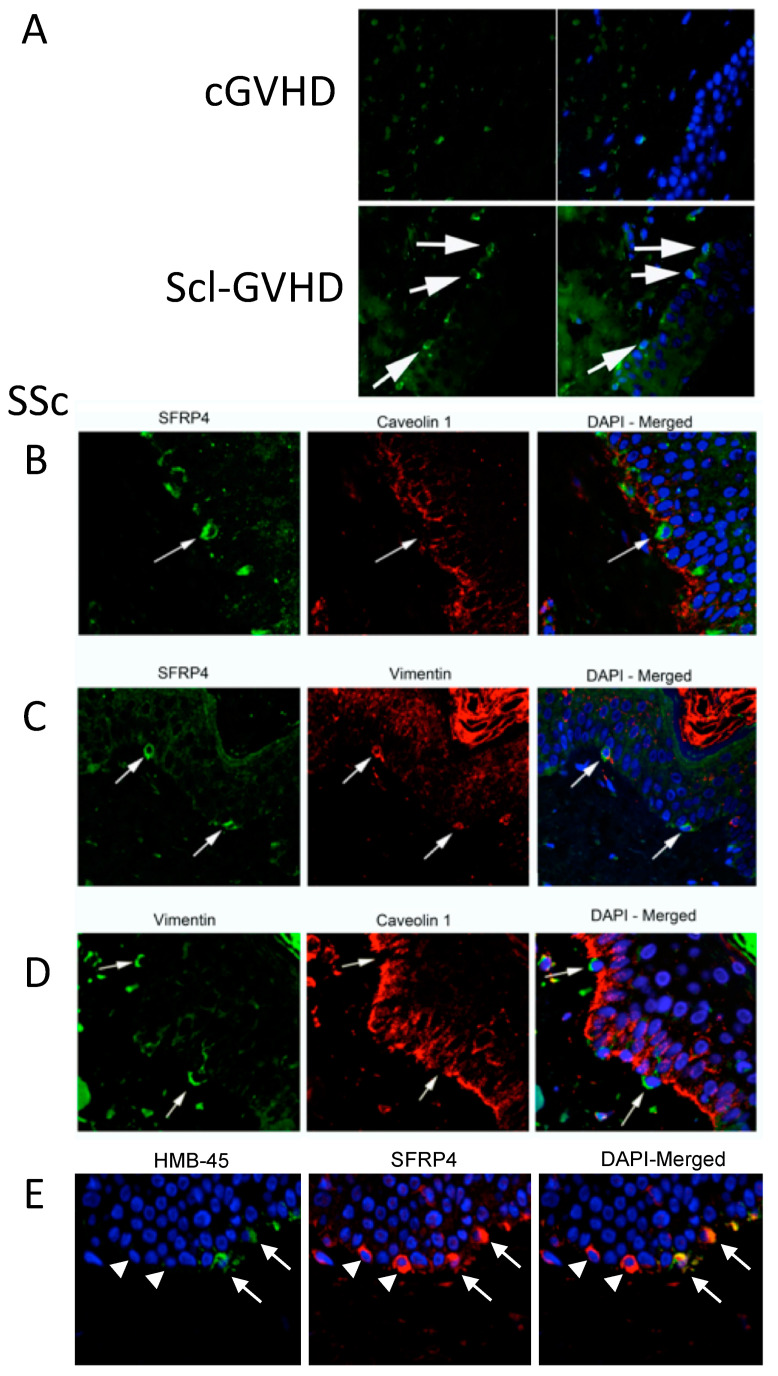
SFRP4 expression is increased in sclGVHD epidermis and the germinal layer. (**A**) Immunofluorescence of SFRP4 in skin samples from a GVHD skin biopsy (cGVHD) and a patient with sclGVHD. The arrows highlight intense SFRP4 positive cells in the basal layer of the epidermis in sclGVHD. Results are representative of 3 subjects (original magnification 400×). (**B**–**E**) Double Immunofluorescence staining of SSc skin biopsies using SFRP4, Caveolin 1, Vimentin and HMB-45 (melanocyte marker). White arrows again highlight the stained positive cells of interest (**B**) SFRP4 and Caveolin 1; (**C**) Vimentin and Caveolin 1; (**D**) Vimentin and Caveolin 1; E; HMB-45 and SFRP4; double stained cells), and the white triangles highlight SFRP4+ and HMB-45-cells (**E**). Original Magnification: (**B**,**C**) = 400×, (**D**,**E**) = 630×.

**Figure 3 jcm-10-05820-f003:**
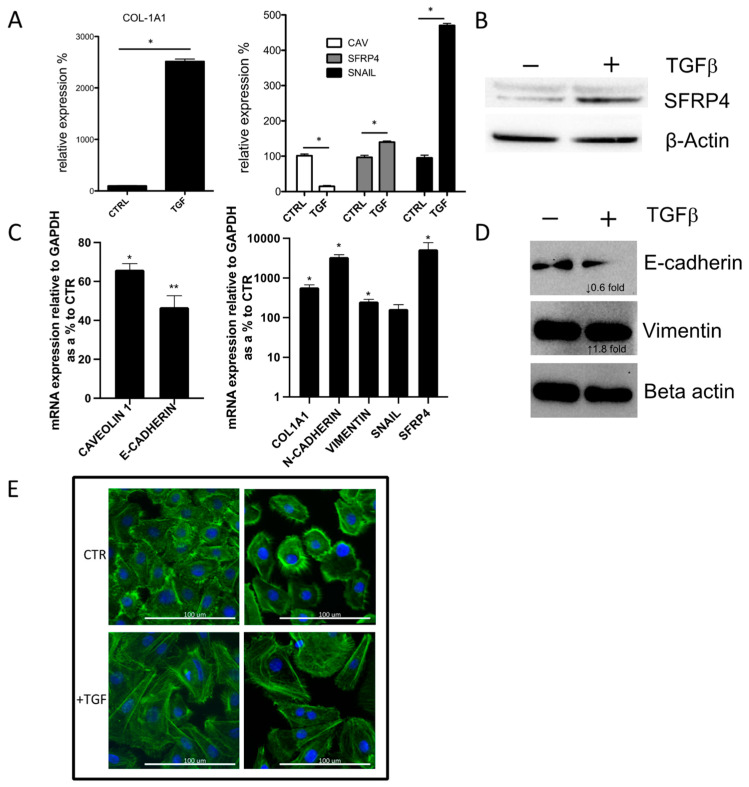
SFRP4 is induced during in vitro TGFβ driven EMT. (**A**) RT-PCR for Col1A1, Caveolin-1, SFRP4 and Snail mRNA levels following 72 h stimulation with 10 ng/mL TGFβ of A549 Epithelial cells. Bars represent mean and standard error (*n =* 3) and represented as a percentage of relative expression to Control (CTRL), normalised by human ribosomal 18 s RNA expression. Statistical significance calculated using paired student t test (* *p <* 0.05, ** *p <* 0.01). (**B**) Western blot of representative experiment for SFRP4 and β-Actin levels in A549 cells before and after 72 h stimulation with TGFβ. (**C**) RT-PCR of the same experiment as above but using primary healthy skin epithelial cells. Bars represent mean and standard error (*n =* 3) and represented as a percentage of relative expression to Control (CTR), normalised by GAPDH expression. (**D**) Western blot of representative experiment for E-cadherin, Vimentin and β-Actin levels in primary healthy skin epithelial cells before and after 72 h stimulation with TGFβ, with the corresponding densitometry calculations for the fold increase in expression between CTR (−) and TGFβ (+) conditions, normalised by β-Actin levels. (**E**) Representative images of phalloidin-stained (green) immunofluorescence of primary healthy skin epithelial cells with and without TGFβ treatment (*n =* 3) (DAPI; blue stained nuclei).

**Figure 4 jcm-10-05820-f004:**
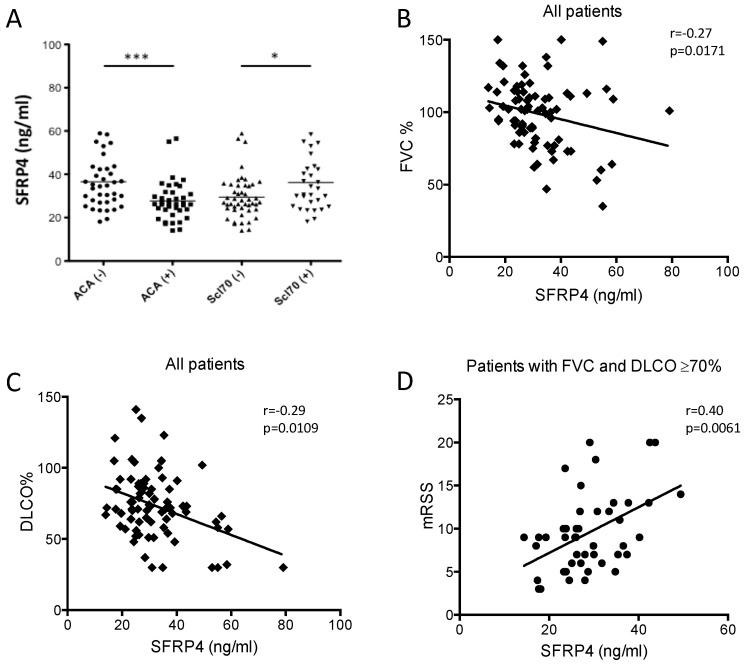
SFRP4 serum concentration as putative biomarker of fibrosis in SSc. (**A**) ACA and Scl-70 autoantibody presence in the cohort of 77 SSc patients (* *p* < 0.05, *** *p* < 0.001). (**B**,**C**) SFRP4 serum level correlation with lung fibrosis in SSc. Spearman R correlation of (**B**) Forced Vital Capacity % predicted levels (FVC%) and (**C**) Diffusion Lung Capacity of carbon monoxide % predicted value (DLCO %) with SFRP4 serum concentration in the cohort of 77 SSc patients. (**D**) Spearman R correlation of SFRP4 concentration with mRSS in 45 patients with FVC% and DLCO% ≥ 70% (r = 0.40; *p* = 0.0061).

**Table 1 jcm-10-05820-t001:** Demographics and clinical characteristics of patients that underwent skin biopsy.

	HC	Scl-GVHD	cGVHD	SSc
N	3	8	5	8
Sex				
F (%)	1 (33)	4 (50)	3 (60)	6 (75)
M (%)	2 (66)	4 (50)	2 (40)	2 (25)
Age (M ± SD)	46.3 ± 12.8	51.3 ± 13.2	48.2 ± 18.1	42.7 ± 9.5
Age of HSTC (M ± SD)	-	48.2 ± 12.5	46.4 ± 17.1	-
HSTC-cGVHD onset (M ± SD)	-	8.7 ± 2.1	7.6 ± 1.6	-
Site biopsy				
Thigh (%)	2 (66)	6 (75)	2 (40)	6 (75)
Forearm (%)	1 (33)	2 (25)	3 (60)	2 (25)
mRSS site biopsy	0	2	0	2
DFS score				
grade 1	3	0	5	0
grade 2	0	1	0	2
grade 3	0	2	0	3
grade 4	0	3	0	2
grade 5	0	2	0	1
Therapy				
CsA (%)		2 (25)	1 (20)	0
MMP (%)	-	7 (87)	4 (80)	1 (12)

HC, healthy controls; SclGVHD, graft versus host disease (fibrotic phenotype); cGVHD, chronic graft versus host disease (without fibrosis); SSc, systemic sclerosis; HSTC, hematopoietic stem cells transplantation; mRSS, modified Rodnan skin score; DFS, dermal fibrosis score; CsA, cyclosporine A; MMP, mycophenolate.

**Table 2 jcm-10-05820-t002:** Epidemiological and clinical features of the 77 SSc patients.

Sex, F/M	62/15
Age, mean (SD), years	57.5 (14.3)
Disease duration, mean (SD), years	15.2 (13.6)
Disease subset, D/L	38/39
ANA +	77 (100 %)
ACA +	39 (50.6%)
Anti-topoisomerase I +	30 (39%)
mRSS, mean (S.D.)/range	9.6 (4.5)/3–20
FVC %, mean (S.D.)/range	98.9 (23.2)/35–150
DLCO %, mean (S.D.)/range	73.4 (23.1)/30–141

SD standard deviation. D diffuse cutaneous SSc; L limited cutaneous SSc; ANA antinuclear antibodies; ACA anticentromere antibodies; mRSS Modified Rodnan skin score; FVC % forced vital capacity % of the predicted value; DLCO % diffusion lung capacity of carbon monoxide % of the predicted value.

## Data Availability

All data shared within manuscript.

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
