# Peer review of "SFRP4 Expression Is Linked to Immune-Driven Fibrotic Conditions, Correlates with Skin and Lung Fibrosis in SSc and a Potential EMT Biomarker"

_jcm, 2021, doi:10.3390/jcm10245820_

Round 1

Reviewer 1 Report

Congratulations on this work and on the improvement of the manuscript. 

Author Response

We would like to thank you for your time and review.

Reviewer 2 Report

The article already underwent multiple rounds of peer review.

It is in my opinion publishable

Author Response

We would like to thank you for your time and your review.

Reviewer 3 Report

The reviewer appreciates the efforts made by the authors. Indeed, the manuscript has been improved in several parts in this revised version. However, there is a point that is of crucial importance to support the evidence of EMT in the epidermis of SSc skin biopsies. This point has not been adequately addressed by the authors. Besides the expression of SFRP4 in epidermal cells detected in skin biopsies, the present set of ex vivo data is not sufficient to demonstrate the presence of EMT in SSc epidermis. As already noted, in the epidermis (especially in the basal layer) vimentin is normally expressed by Langerhans cells, and the few vimentin-positive cells shown by the authors are all located in the basal layer. Moreover, the only one vimentin-positive cell displayed in figure 2D (pointed by white arrow) has the characteristic morphology of Langerhans cells with an apical process spreading toward the more superficial epidermal layers. Therefore, the only possibility to discern whether these cells are normally resident Langerhans cells or fibroblast-like cells derived from keratinocytes is to perform a double immunostaining for vimentin and CD207 (langerin). Otherwise, the authors could perform double staining for SFRP4 and another marker of activated fibroblasts/myofibroblasts (e.g. α-SMA) so that, if they detect double stained cells in the epidermis, it will be reliable to propose the occurrence of EMT in SSc skin and the involvement of SFRP4 in this process.

Author Response

Thank you for your time and for reviewing our manuscript. We have addressed your concerns in the cover letter and amended the manuscript accordingly. 

Round 2

Reviewer 3 Report

No further comments.

This manuscript is a resubmission of an earlier submission. The following is a list of the peer review reports and author responses from that submission.

Round 1

Reviewer 1 Report

In this manuscript the authors investigated the involvement of SFRP4, a molecule that has been already reported to be increased in SSc and in an animal model of skin fibrosis, in the pathogenesis of SSc fibrosis. In particular, the authors employed sclGVHD and cGVHD skin biopsies to validate the importance of SFRP4 in the profibrotic process and then identified, in the basal layer of SSc epidermis, cells positive for both SFRP4 and vimentin and negative for caveolin-1. With additional experiments on A549 epithelial cells they tested the hypothesis that SFRP4 and vimentin expression in the epidermis of SSc skin biopsies could represent a marker of EMT. Indeed, they demostrated that SFRP4 is induced during in vitro TGFβ driven EMT. Moreover, as circulating SFRP4 correlated with skin and lung fibrosis in SSc, they also suggested that serum SRFP4 concentrations might function as biomarker of fibrogenesis. Although of potential interest, the experiments performed in this study are not sufficient to fully elucidate the involvement of SFRP4 in the EMT process. For this reason, some major issues have to be addressed:

-It is not clear, in the abstract, the number of SSc skin biopsies and SSc sera that have been analyzed. What does the number 85 refer to? Please specify. Moreover, the number of healthy volunteers that have been used as controls is very small and could be a limitation of the study.

-In SSc skin tissue, SFRP4 positive cells were detected in the basal layer of the epidermis; these cells also expressed vimentin. Thus, the authors conclude that these cells may be epithelial cells that undergo EMT. However, an important issue has to be considered: the basal layer is also home to Langerhans cells. These cells are of mesenchimal origin and are known to express vimentin. Are the authors sure that SFRP4/vimentin positive cells are not Langerhans cells? A double immunofluorescence staining with the CD207 (langerin) marker has to be performed to exclude SFRP4 positivity of Langerhans cells.

-In figures and legends some data are missing. In figure 2, images of skin from healthy controls are not shown. Moreover, what do the arrowheads represent? In figure 3A, the increase in snail mRNA is not represented (it is in figure 3B), moreover statistical significance is missing.

-the experiments performed on A549 cells are not sufficient to test the hypothesis that SFRP4 and vimentin expression could represent a marker of EMT. First, the authors showed only an increase in mesenchymal markers (Col1A1 and snail) but not a decrease in epithelial markers (such as E-cadherin). Moreover, the increase/decrease of mRNA expression of such markers should be demonstrated also at a protein level. As already reported for SFRP4, the expression of vimentin in cells challenged with TGFβ should be investigated. To fully demonstrate the presence of cells that are undergoing the EMT process, some phase contrast images representing changes in cell morphology should be displayed. Moreover, double immunofluorescence staining (with mesenchymal and epithelial markers) is necessary to show cells in transition.

- No hypothesis was made in the discussion on how SFRP4 can induce or be involved in EMT in SSc.

Author Response

We thank the reviewer for taking the time to review our study. Please see our responses below:

-It is not clear, in the abstract, the number of SSc skin biopsies and SSc sera that have been analyzed. What does the number 85 refer to? Please specify. Moreover, the number of healthy volunteers that have been used as controls is very small and could be a limitation of the study.

85 referred to the total skin biopsies and sera for ssc. We thank the reviewer for highlighting the confusion and have amended the abstract accordingly:

‘To determine its role in immune-driven fibrosis, we analysed SSc and sclerotic Chronic Graft Versus Host Disease (sclGVHD) biosamples; skin biopsies (n=24) from chronic GVHD patients (8 with and 5 without sclGVHD), 8 from SSc and 3 healthy controls (HC) were analysed by immunofluorescence (IF) and SSc patient sera (n=77) assessed by ELISA.’ Lines 17-19

-In SSc skin tissue, SFRP4 positive cells were detected in the basal layer of the epidermis; these cells also expressed vimentin. Thus, the authors conclude that these cells may be epithelial cells that undergo EMT. However, an important issue has to be considered: the basal layer is also home to Langerhans cells. These cells are of mesenchimal origin and are known to express vimentin. Are the authors sure that SFRP4/vimentin positive cells are not Langerhans cells? A double immunofluorescence staining with the CD207 (langerin) marker has to be performed to exclude SFRP4 positivity of Langerhans cells.

We thank the reviewer for this valuable comment and that our experimental data do not exclude that some of these positive cells may be langherans cells. Several papers have shown increased expression of SFRP4 in SSc fibroblasts, and both our in vitro and biomarker data do suggest the importance of the secreted factor in SSc. Nevertheless for accuracy we have added the following sentence in the discussion

‘Although the in vitro and biomarker data do suggest that TGF may induce SFRP4 in epithelial cells, another possible origin of SFPR4 staining in SSc may be langherans cells. While their relative amount would not justify an increase in serum concentration detectable in SSc patients, further experiments should be designed to assess this possibility.’

-In figures and legends some data are missing. In figure 2, images of skin from healthy controls are not shown. Moreover, what do the arrowheads represent? In figure 3A, the increase in snail mRNA is not represented (it is in figure 3B), moreover statistical significance is missing.

Figure 2: Healthy controls were not included as we used cGVHD (no fibrosis skin model) to show no SFRP4 expression. The arrowheads are explained in the figure legend, however, we agree with the reviewer that these could be explained better. Figure legend now updated to:

‘White arrows again highlight the stained positive cells of interest (B; SFRP4 and Caveolin 1; C; Vimentin and Caveolin 1; D Vimentin and Caveolin 1; E; HMB-45 and SFRP4; double stained cells), and the white triangles highlight SFRP4+ and HMB-45- cells (E). Original Magnification: B,C=400X, D,E=630X.’ lines 232-236.

Figure 3: We have amended the labelling so snail and col1a1 included together and referred to correctly in the text now. We have included the statistical significance into the figure legends and figures as suggested.

-the experiments performed on A549 cells are not sufficient to test the hypothesis that SFRP4 and vimentin expression could represent a marker of EMT. First, the authors showed only an increase in mesenchymal markers (Col1A1 and snail) but not a decrease in epithelial markers (such as E-cadherin). Moreover, the increase/decrease of mRNA expression of such markers should be demonstrated also at a protein level. As already reported for SFRP4, the expression of vimentin in cells challenged with TGFβ should be investigated. To fully demonstrate the presence of cells that are undergoing the EMT process, some phase contrast images representing changes in cell morphology should be displayed. Moreover, double immunofluorescence staining (with mesenchymal and epithelial markers) is necessary to show cells in transition.

To address these concerns, we performed additional EMT studies using primary skin keratinocytes to build on the A549 data (which is a carcinoma human cell line from lung). We now show that TGF can induce EMT also in this cell line (added to fig 3 C-E). To address the concerns for additional EMT markers, we included N-cadherin, vimentin and e-cadherin. Furthermore, we have performed WB analysis for protein expression of e-cadherin and vimentin. We have also included cell morphology images looking at phalloidin. Altogether we feel these revisions address the concerns of the reviewer. We have included the additional methods, altered the figure legend, results and discussion accordingly.

- No hypothesis was made in the discussion on how SFRP4 can induce or be involved in EMT in SSc.

We have included in the discussion:

Further studies are needed to determine the putative role of SFRP4 in EMT and whether this is involved in myofibroblast differentiation, since previously SFRP4 has been shown to be a marker of this subset of SSc fibroblasts [32]. Previous observations show strong correlation between SFRP4 and other Wnt-related gene expression, including WIF1 and WNT2, in SSc skin and fibrosis [15, 16, 32, 33], thus it is likely that SFRP4 is a marker for aberrant Wnt signalling, and warrants further research. Additional studies are needed to determine whether SFRP4 and EMT is a pathologic step in fibrosis of SSc and sclGVHD.

Reviewer 2 Report

An interesting original study showing that Secreted Frizzled Receptor Protein 4 expression was increased during 23 Tumor Growth Factor b-induced Epithelial-Mesenchymal-Transition, and was correlated with the severity of lung and skin fibrosis in SSc, possibly functioning as a biomarker of transition of cells from the epithelial to the mesenchymal type. I have some minor queries:

Various skin biopsies have been performed during this study, so a generical ethical committee approval does not suffice.

Please also provide the number of the approval for the study, the year of approval, and the complete name of the ethical committee.

Thank You

Author Response

Many thanks for your comments and feedback. We have addressed your comment below:

Various skin biopsies have been performed during this study, so a generical ethical committee approval does not suffice. Please also provide the number of the approval for the study, the year of approval, and the complete name of the ethical committee.

We have updated the methods and included an ethics statement:

Ethic statement: Study was approved by Ethical Committee of University of Verona, Italy (Study number VR409).

Reviewer 3 Report

I consider the study interesting, well designed, and useful.

I would modify the structure. May be Results section should show just results, and transfer to discussion and introduction, comments and authors deductions.

Some examples of sentences that are not results:

-as described previously (14).

-Immunofluorescence studies confirmed previously described increased SFRP4 expression in SSc (data not shown) [4, 10-17].

-we analysed the expression of these two markers in vitro employing a well-established model of EMT [29]

-Since SFRP4 is a soluble protein and because of its increased expression in the skin, we tested the hypothesis that its concentration may reflect the amount of fibrogenesis in SSc patients.

- To test the hypothesis that SFRP4 and Vimentin expression in the epidermis of SSc skin biopsies could represent a marker of EMT, we analysed the expression of these two markers in vitro employing a well-established model of EMT [29].

Author Response

I would modify the structure. May be Results section should show just results, and transfer to discussion and introduction, comments and authors deductions.

Some examples of sentences that are not results:

-as described previously (14).

-Immunofluorescence studies confirmed previously described increased SFRP4 expression in SSc (data not shown) [4, 10-17].

-we analysed the expression of these two markers in vitro employing a well-established model of EMT [29]

-Since SFRP4 is a soluble protein and because of its increased expression in the skin, we tested the hypothesis that its concentration may reflect the amount of fibrogenesis in SSc patients.

Many thanks to your review of our study. Please find our response to your report:

- To test the hypothesis that SFRP4 and Vimentin expression in the epidermis of SSc skin biopsies could represent a marker of EMT, we analysed the expression of these two markers in vitro employing a well-established model of EMT [29].

Many thanks to the reviewer for highlighting this. We have amended and adapted/moved the sentences as suggested.